# Evolve to Adapt, Not Guess: A Gradient-Free and Robust Framework for Layer-Wise Fine-Tuning via Evolutionary Learning Rate Optimization

## Abstract

Fine-tuning pretrained neural networks for domain adaptation requires careful adjustment of layer-specific learning rates, yet existing strategies often rely on manual heuristics or global schedules that fail to capture diverse adaptation patterns. This challenge is amplified in data-scarce settings, where gradient-based hyperparameter optimization suffers from high variance. To this end, we present REVO-Tune, which systematically employs evolutionary optimization to discover optimal layer-specific learning rates during fine-tuning neural networks automatically. Our approach introduces two encoding strategies: binary representation that selectively adapts layers with a shared global rate for computational efficiency, and continuous representation that assigns per-layer learning rates for fine-grained control. Both strategies use gradient-free population-based search to explore optimal configurations. Across diverse datasets and architectures, REVO-Tune consistently improves fine-tuning performance, yielding 2-4% higher accuracy and 1-3% higher AUC than standard fine-tuning approaches. The continuous encoding excels in performance-critical scenarios, while binary encoding offers substantial efficiency-accuracy trade-offs. Our empirical analysis demonstrates that evolutionary optimization can effectively complement modern adaptive optimizers, providing practical improvements for automated fine-tuning in resource-constrained environments where manual hyperparameter tuning is impractical. Code is provided as supplementary material.

## 1 Introduction

Deep neural networks (DNNs) demonstrate exceptional performance across domains through their ability to learn hierarchical representations from raw data Krizhevsky et al. (2017); LeCun et al. (2015). However, DNNs suffer performance deterioration when confronted with distribution shifts between source pretraining and target datasets Recht et al. (2019); Hendrycks et al. (2019); Koh et al. (2021). This phenomenon occurs due to statistical disparities from changes in data collection methods, environmental variations, or demographic differences, leading to mismatched representations and reduced generalization Hendrycks & Dietterich (2019); Koh et al. (2021). For instance, DNNs trained on high-quality studio images struggle with real-world images containing noise, blur, or varying lighting conditions Saenko et al. (2010).

Addressing distribution shifts is crucial for developing robust DNN models that perform consistently across scenarios. Researchers have proposed strategies to mitigate these shifts, including transfer learning (domain adaptation) He et al. (2024), domain generalization (data augmentation and meta-learning) Ding et al. (2022); Liu et al. (2022b), and unsupervised approaches (self-supervised learning and unsupervised domain adaptation) Dragoi et al. (2022); Dasgupta et al. (2022). These strategies align source and target distributions by learning invariant representations, simulating varied scenarios, or imposing consistency constraints Tzeng et al. (2017).

Fine-tuning pre-trained models on limited labeled target datasets has emerged as an effective adaptation strategy that outperforms domain generalization and unsupervised approaches Kirichenko et al.

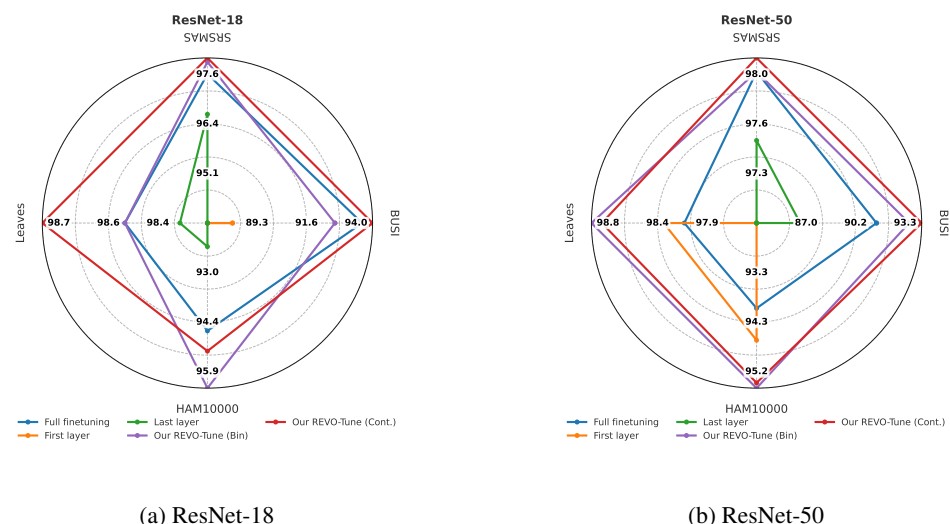

(a) ResNet-18                              (b) ResNet-50

Figure 1: Motivation for layer-wise learning rate optimization. Radar plots compare five fine-tuning strategies across four diverse datasets (BUSI, HAM10000, Leaves, SRSMAS) at 40% training data using ResNet-18 and ResNet-50. Radii represent normalized AUC performance

(2022). This technique adapts models with general representations from large source datasets to smaller, task-specific datasets, leveraging transferable knowledge while customizing components for the target domain Lee et al. (2022). Fine-tuning efficacy depends on balancing adaptation to novel data while preserving valuable pre-trained knowledge, mitigating overfitting while enabling adjustment to domain-specific characteristics Kumar et al. (2022).

Consequently, several techniques have been developed to optimize the trade-off between adaptation and preservation of pre-trained knowledge. The most common approach uses lower learning rates than during pre-training, enabling gradual adaptation Kornblith et al. (2019); Li et al. (2020). Another strategy selectively freezes and gradually unfreezes layers from top to bottom, preserving low-level features while allowing higher-level adaptation Howard & Ruder (2018); Mukherjee & Awadallah (2019). Researchers have also explored assigning different learning rates to layers—higher rates for upper and lower rates for bottom layers—facilitating rapid high-level adaptation while maintaining stable low-level features Ro & Choi (2021); Shen et al. (2021).

Although existing fine-tuning methods show promising results, they have critical limitations. Lower learning rates may cause slow convergence, requiring more computational resources and leading to poor adaptation Smith & Topin (2019). Selectively freezing and unfreezing layers is susceptible to hyperparameter tuning, where inappropriate decisions significantly degrade performance Lee et al. (2022). Setting different learning rates requires tuning multiple hyperparameters, causing complexity and training instability. Improperly calibrated rates may change layers too rapidly, overriding valuable features. Additionally, some techniques require trial-and-error methods and gradient information that may not be consistently available or computationally feasible Choi et al. (2024).

To address these limitations, we present *REVO-Tune*, which applies evolutionary optimization to automatically discover optimal layer-specific learning rates during fine-tuning. Our approach is motivated by the observation that different layers may require distinct learning rates to adapt effectively to target tasks. REVO-Tune uses gradient-free population-based search to explore layer-wise learning rate configurations, making it particularly suitable for data-scarce scenarios where gradient-based hyperparameter optimization may be unreliable due to high variance. The method systematically searches for learning rate patterns that balance adaptation to target data while preserving valuable pretrained knowledge.

REVO-Tune approach addresses the challenge of representing layer-wise learning rate configurations for evolutionary optimization through two encoding strategies. The *binary encoding* selectively adapts layers using a shared global learning rate for computational efficiency, while the *continuous*

*encoding* assigns individual learning rates to each layer for fine-grained control. Both strategies use mixed variable CMA-ES to evolve populations of candidate solutions over successive generations through fitness evaluation, selection, and adaptation operations.

Figure 1 illustrates that existing fine-tuning strategies exhibit inconsistent performance across target domains. While full fine-tuning may excel on one dataset, it underperforms significantly on others; similarly, layer-specific approaches such as first-layer and last-layer fine-tuning demonstrate dataset-dependent effectiveness. This performance variability motivates the need for adaptive layer-wise learning rate optimization. Our proposed REVO-Tune method addresses this limitation by maintaining consistently high performance across all evaluated datasets through evolutionary optimization for automated layer-specific learning rate discovery.

Our key technical contributions and breakthroughs in this work include the followings:

- We employ CMA-ES evolutionary optimization to automatically discover layer-specific learning rates for neural network fine-tuning, providing an alternative to manual hyperparameter search.

- We develop and evaluate binary and continuous encoding strategies for representing layer-wise learning rate configurations in evolutionary optimization for fine-tuning tasks.

- Extensive experiments were conducted on multiple datasets and models to validate the efficacy of the proposed approach. Results demonstrate that the proposed method outperforms existing fine-tuning techniques, underscoring its versatility and robustness.

- We show that evolutionary optimization is particularly effective in data-scarce scenarios and analyze the discovered layer-wise learning rate patterns across different datasets and training conditions.

## 2 RELATED WORKS

Recent advancements in fine-tuning strategies for DNNs have highlighted the effectiveness of differential learning rates across layers. Discriminative fine-tuning Howard & Ruder (2018) applies lower learning rates to early layers to preserve generalizable features while allowing greater adaptation in later layers. Several approaches have refined layerwise adjustments, including cyclical learning rates Smith (2017) and the Lookahead optimizer Zhang et al. (2019), which enables granular modifications through parameter space exploration. Automated approaches have gained attention, with AutoLR Ro & Choi (2021) combining layerwise pruning with automatic tuning, while other methods initialize layerwise rates using gradient magnitudes to enhance stability Lee et al. (2022).

Recent advancements emphasize the importance of tailoring fine-tuning processes. The Fisher Information Matrix has been used to identify crucial adaptation layers for evolving data streams Park et al. (2024), while evolutionary search techniques for selective layer freezing have achieved state-of-the-art few-shot learning Shen et al. (2021). However, this method Shen et al. (2021) searches among $K$ candidate learning rates for $M$ layers, requiring $M \log K$ variables using binary encoding, which can be computationally intensive. While these layerwise learning strategies have advanced deep learning, they face limitations including slower convergence, hyperparameter sensitivity, gradient dependency, and increased computational complexity.

Fine-tuning DNNs requires recognizing the unequal contribution of individual layers to model performance. This uneven distribution significantly impacts overall accuracy Lee et al. (2022), with some layers contributing substantially while others exert minimal influence—a critical consideration often overlooked in fine-tuning. Experimental evidence Kaplun et al. (2023) demonstrates that layer importance cannot be reliably predicted from structural properties such as depth, parameter count, or spatial resolution. The same architecture can exhibit dramatically different fine-tuning profiles across tasks or initializations, revealing non-linear relationships between layer properties and performance Kaplun et al. (2023). Simplistic assumptions about architectural characteristics frequently lead to suboptimal fine-tuning strategies.

This complexity Lee et al. (2022); Kaplun et al. (2023) necessitates careful analysis of specific fine-tuning profiles for each architecture-dataset combination. Effective fine-tuning must balance adaptation to target characteristics while preserving valuable pre-trained representations. Evolutionary optimization offers an automated alternative to manual tuning, efficiently exploring the search space

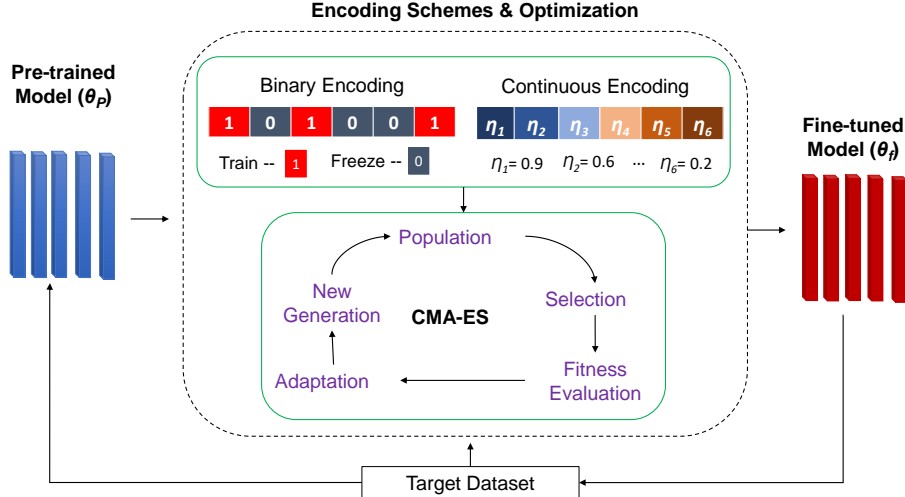

Figure 2: The overall framework of the proposed method. First, we initialize the model with the pre-trained weights $\theta_p$, which are obtained from training on a large-scale generic dataset such as ImageNet Deng et al. (2009). Then, the evolutionary optimization process is initiated using one of two different encoding strategies. During the optimization process, the model is fine-tuned using the corresponding update rule with the generated solution and evaluated on the validation set. After the optimization, the best-performing genome is selected as the final learning rate representation, and the resulting model weights are set as the final weights for evaluation on the test set.

of fine-tuning strategies through iterative evolution of candidate solutions. These algorithms discover optimal configurations that balance model adaptation with knowledge retention through systematic mutation and recombination.

## 3 OUR APPROACH: REVO-TUNE

This section outlines the problem statement and presents the proposed method that employs evolutionary search to optimize layer-specific learning rates for neural network fine-tuning.

### 3.1 OPTIMIZATION PROBLEM

The problem setting involves two datasets with distinct distributions: a larger dataset that follows the source distribution $P_{\text{src}}$, and a relatively smaller dataset $D_{\text{tgt}}$ that adheres to the target distribution $P_{\text{tgt}}$. The primary objective is to achieve high accuracy on the target data by utilizing the related yet distinct source distribution. This challenge is common in real-world applications requiring adaptability across varying data distributions.

First, a network is pre-trained to minimize the loss on the source dataset, resulting in the model $\theta_p$, which has high accuracy on the source distribution. Next, a fine-tuning stage is performed, starting from the pretrained model parameters and minimizing the loss on the labeled target data $D_{\text{tgt}}$, resulting in the model $\theta_f$. The proposed method employs an evolutionary search to optimize layerwise learning rates and enhance the fine-tuning performance of the model. This approach employs two encoding strategies to facilitate layer-specific adjustments during the evolutionary optimization process using the loss on validation set as an objective $\mathcal{L}_{\text{tgt}}(D_{\text{tgt}}, \theta_f) = \mathbb{E}_{(x,y) \sim P_{\text{tgt}}}[l((x, y), \theta_f)]$. Finally, the performance of the fine-tuned model $\theta_f$ is evaluated on held-out data from the target distribution.

## 3.2 OVERALL FRAMEWORK

This section presents the proposed method for enhancing DNN fine-tuning through evolutionary search. The core concept involves optimizing layer-specific learning rates, enabling more precise control than using a single global learning rate across all layers. Fig. 2 depicts the overall framework. First, pretrained model weights $\theta_p$ are obtained from a large-scale dataset such as ImageNet Deng et al. (2009). Next, evolutionary optimization is initiated with a population where each genome represents layer configurations, with genes encoding either layer selection (binary) or learning rate values (continuous). The method employs two encoding schemes for layer-specific control: *binary and continuous representation*.

After initialization, the fitness of each genome is evaluated. The DNN with pretrained weights is initialized, then the encoded layer configuration is applied, involving either freezing/fine-tuning layers (binary representation) or setting layer-specific learning rates (continuous representation). The model is fine-tuned using the generated solution and corresponding update rule on a portion of the target dataset with the configured layer-wise settings. Performance is then evaluated on a held-out validation set from the target dataset, serving as the fitness score for each genome. This evaluation enables the evolutionary algorithm to assess the effectiveness of each encoded layer configuration and estimate how well it adapts to the target task. This approach allows robust comparison of various layer configurations and their influence on model performance.

The fitness score guides the evolutionary search, enabling the algorithm to iteratively refine layer configurations and discover the optimal setup for fine-tuning on the target task. After optimization, the best-performing genome is selected as the final learning rate representation, and the resulting model weights are used for evaluation on the test set. Consistent evaluation across binary and continuous representation strategies ensures fair comparison of the two methods.

## 3.3 ENCODING STRATEGIES

The encoding strategy is crucial in evolutionary optimization because it defines the representation of candidate solutions as individuals in the population. These strategies significantly affect the efficiency, effectiveness, and applicability of the algorithm across optimization problems. The proposed method employs two distinct encoding strategies for layer-specific control in the evolutionary process: *binary and continuous representation*.

### 3.3.1 BINARY REPRESENTATION

The first approach uses binary representation for the genome in the evolutionary algorithm population. This representation encodes configurations for selectively freezing or fine-tuning individual layers. Each genome corresponds to a unique configuration, with each gene representing a specific DNN layer. The gene holds a binary value of 0 or 1, as depicted in Fig. 2. A value of 0 designates that layer weights remain frozen during fine-tuning, preserving pretrained knowledge, while a value of 1 indicates the layer is updated and fine-tuned to the target task.

All layers marked for fine-tuning (gene value of 1) in this binary representation strategy share a single, fixed learning rate. This simplifies the learning rate selection process by requiring the evolutionary algorithm to optimize only one global learning rate for fine-tuned layers rather than individual rates for each layer. The binary strategy balances flexibility and efficiency by identifying the optimal layer subset to adapt using this simplified scheme. The evolutionary process can pinpoint critical layers for fine-tuning without the complexity of optimizing unique learning rates per layer.

### 3.3.2 CONTINUOUS REPRESENTATION

In continuous representation, each genome is structured as a vector where each gene directly encodes the learning rate of the corresponding model layer. Learning rates are represented as continuous values in [0, 1]. Values closer to 0 indicate slower adaptation during fine-tuning, allowing gradual changes to preserve pretrained knowledge. Values closer to 1 allow faster weight modification, facilitating rapid adaptation to the target task.

This continuous representation provides fine-grained control over the fine-tuning process. The evolutionary algorithm can explore a broader range of configurations and find optimal adaptation

speeds across network layers by assigning tailored learning rates to each layer. This control is beneficial for complex neural network architectures where layers have different roles and sensitivities to fine-tuning. By tuning learning rates at the individual layer level, the continuous representation strategy better accommodates the unique characteristics of each layer, more effectively fine-tuning the model on the target task.

### 3.4 Weight Update Rule

The weight update rule outlines the systematic process by which network parameters are iteratively adjusted as the model learns from a dataset. The weight update rule for both encoding schemes is as follows:

*Binary Representation*: For a layer $l$, let $b_l \in \{0, 1\}$ be the binary value in the genome, with 0 indicating a frozen layer and 1 a fine-tuned layer. Let $\eta$ be the global learning rate. The weight update rule is:

$$\theta_l^{t+1} = \theta_l^t - b_l * \eta * \nabla_{\theta_l} L \tag{1}$$

*Continuous Representation*: Here, each layer $l$ has an associated learning rate $\eta_l \in [0, 1]$ encoded within the genome. The weight update rule is:

$$\theta_l^{t+1} = \theta_l^t - \eta_l * \nabla_{\theta_l} L \tag{2}$$

In both cases, $\nabla_{\theta_l} L$ represents the gradient of the loss function concerning layer weights $l$, computed using standard backpropagation. Evolutionary optimization applies nature-inspired techniques, such as evaluation, selection, and adaptation, in the proposed approach. This evolutionary process continues to discover highly effective layerwise fine-tuning strategies over multiple generations.

### 3.5 Evolutionary Optimization Method

We employ mixed-variable CMA-ES Uchida et al. (2024), a derivative-free optimization algorithm designed for mixed discrete-continuous problems, to determine layer-specific learning rates through progressive adaptation of a multivariate normal distribution:

1. *Initialization*: Candidate solutions are sampled from a multivariate normal distribution parameterized by a mean vector and covariance matrix.
2. *Evaluation*: Each candidate undergoes fitness assessment on a validation set, measuring fine-tuning performance improvement.
3. *Selection*: Mixed-variable CMA-ES selects best-performing candidates based on fitness scores.
4. *Adaptation*: The distribution's covariance matrix and mean adapt based on selected solutions, focusing search on promising regions.
5. *Sampling*: New candidates are sampled and evaluated iteratively until convergence.

For binary encoding, mixed-variable CMA-ES optimizes discrete layer selections and continuous global learning rates. For continuous encoding, it optimizes per-layer learning rates while maintaining covariance adaptation properties for mixed variable types.

### 3.6 Some Advantages in Data-Scarce Scenarios

Traditional gradient-based fine-tuning methods require extensive labeled data from the target domain to compute accurate gradient estimates. When data are limited, these gradient calculations become unreliable due to increased noise, hindering optimization and leading to suboptimal performance. The proposed REVO-Tune approach, employing CMA-ES for layerwise learning rate optimization, offers significant advantages in low-data regimes. CMA-ES operates in a black-box manner, relying solely on final performance metrics (e.g., validation accuracy) to guide optimization. This eliminates the need for explicit gradient computations and associated susceptibility to noise in data-scarce scenarios.

We let $\mathcal{L}(D_{\text{tgt}}, \theta)$ represent the model performance metric on the validation set, with $\theta$ denoting the entire set of model weights. The goal of CMA-ES is to optimize this objective function: minimize $\mathcal{L}(D_{\text{tgt}}, \theta)$. By directly focusing on the evolutionary search for optimal model performance, our CMA-ES-based method effectively navigates fine-tuning even when labeled target data is limited.

Black-box, gradient-free optimization approaches like CMA-ES offer several advantages in data-scarce scenarios. First, they have reduced sensitivity to noise that emerges when gradient estimates are computed with limited data, allowing the optimization process to filter out misleading noisy gradients. Second, black-box methods demonstrate applicability to non-smooth or non-differentiable objective functions where gradient-based methods may struggle.

## 4 EXPERIMENTS

This section details our experimental methodology, encompassing dataset characteristics, implementation specifications, and a comprehensive analysis of empirical results.

### 4.1 EXPERIMENTAL SETUP

We evaluated the proposed approach on four diverse datasets: Structure Rosenstiel School of Marine and Atmospheric Science (SRSMAS) Gómez-Ríos et al. (2019), a dataset of coral reef types; Breast Ultrasound Images (BUSI) Al-Dhabyani et al. (2020); Human Against Machine with 10000 training images (HAM10000)Tschandl et al. (2018), a dermatoscopic image dataset of skin lesions; and Leaves Rauf et al. (2019), a dataset of citrus leave diseases. These datasets span different applications, including underwater imaging, medical imaging, and plant classification, allowing the assessment of the generalizability of the proposed method across diverse domains.

We conducted experiments using two well-known DNN architectures: ResNet-18 and ResNet-50 He et al. (2016). These models were trained on the ImageNet Deng et al. (2009) dataset and served to initialize the fine-tuning experiments. Their performance was evaluated using the area under the receiver operating characteristic curve (AUC) and the overall classification accuracy. The AUC provides a comprehensive measure of the trade-off between the true- and false-positive rates, whereas accuracy quantifies the overall correctness of the predictions. Further, we compared the proposed method against several baselines: full fine-tuning, first-layer fine-tuning, last-layer fine-tuning, a relative gradient norm (auto-RGN) Lee et al. (2022) that sets the learning rates based on the gradient norm of the layers, and a set encoding-based Shen et al. (2021) evolutionary algorithm (a genetic algorithm approach that optimizes a set of learning rates for each layer).

In our implementation, we use CMA-MV with a population size of 10 and run optimization for 10 iterations, providing a computational budget of 100 evaluations. For binary encoding, the algorithm jointly optimizes binary layer selections and a global learning rate of 1e-3. For continuous encoding, it optimizes individual learning rates for each layer in the range [0,1]. These parameters provide an effective balance between exploration and computational efficiency for the layer-wise optimization task. For the gradient-based baselines, we tuned the learning rate and weight decay using the HyperOpt Bergstra et al. (2013) search algorithm with successive halving, as implemented in the Ray Tuner Liaw et al. (2018) library, with the same computational budget as the evolutionary algorithms. Other hyperparameters, such as the batch size, are consistent across all methods and datasets, following the recommended settings in the original papers for the respective models.

### 4.2 EXPERIMENTAL RESULTS AND ANALYSIS

Tables 1 - 2 present the experimental results obtained by comparing the proposed REVO-Tune approach with two strategies (binary and continuous representation) against the baseline schemes. The experimental results reveal that the proposed REVO-Tune using the continuous representation strategy consistently outperforms the baselines (i.e., full fine-tuning, first-layer fine-tuning, last-layer fine-tuning, auto-RGN Lee et al. (2022), and set encoding Shen et al. (2021)) across all four datasets (BUSI, HAM10000, Leaves, and SRSMAS) with varying training data sizes.

REVO-Tune with binary representation also displays strong performance, often surpassing baselines and occasionally matching continuous representation results, indicating that even simplified layer

Table 1: ResNet–18 results across BUSI Al-Dhabyani et al. (2020), HAM10000 Tschandl et al. (2018), Leaves Rauf et al. (2019), and SRSMAS Gómez-Ríos et al. (2019). For REVO–Tune we report mean ± std over 3 runs. Baselines include full, first, and last layer fine-tuning, Auto-RGN Lee et al. (2022), and Set encoding Shen et al. (2021). Best per column is in **bold**.

| Data size | Method | BUSI AUC | BUSI Acc | HAM10000 AUC | HAM10000 Acc | Leaves AUC | Leaves Acc | SRSMAS AUC | SRSMAS Acc |
|---|---|---|---|---|---|---|---|---|---|
| 0.05 | Full finetuning | 81.2 | 67.0 | 86.0 | 73.4 | 96.1 | 82.3 | 86.8 | 42.2 |
| | First layer | 59.0 | 55.0 | 85.4 | 72.0 | 94.4 | 74.7 | 74.0 | 18.5 |
| | Last layer | 67.6 | 56.2 | 86.8 | 71.5 | 94.2 | 71.6 | 80.3 | 20.5 |
| | Auto-RGN | 73.4 | 61.2 | 86.3 | 72.5 | 93.7 | 75.1 | 81.9 | 25.3 |
| | Set encoding | **86.7** | 70.9 | 84.9 | 75.1 | 96.2 | 79.3 | **87.0** | **49.4** |
| | REVO-Tune (Bin) | 81.8±0.5 | 67.5±1.9 | 86.5±0.5 | 74.5±0.2 | 96.5±1.8 | 83.6±2.1 | 84.3±0.8 | 44.2±1.1 |
| | REVO-Tune (Cont.) | 86.1±1.5 | **72.3±0.9** | **89.1±0.4** | **76.8±0.2** | **97.2±0.1** | **84.9±0.4** | **87.0±0.9** | 39.6±1.6 |
| 0.2 | Full finetuning | 92.4 | 80.8 | 92.7 | 80.7 | 97.7 | 87.5 | 95.4 | 59.5 |
| | First layer | 83.4 | 65.2 | 92.5 | 78.3 | 97.6 | 84.7 | 94.0 | 56.3 |
| | Last layer | 84.8 | 70.7 | 90.5 | 74.7 | 97.4 | 84.2 | 94.1 | 49.0 |
| | Auto-RGN | 87.2 | 74.1 | 92.5 | 78.0 | 97.1 | 82.3 | 94.5 | 61.5 |
| | Set encoding | **92.7** | **82.1** | 93.3 | 81.0 | 98.4 | 89.9 | 95.9 | **72.9** |
| | REVO-Tune (Bin) | 90.4±0.2 | 79.3±0.7 | 93.4±1.1 | 80.6±0.9 | **98.7±0.5** | 88.0±1.7 | 96.5±0.5 | 71.3±2.2 |
| | REVO-Tune (Cont.) | 92.0±3.9 | 80.1±5.5 | **94.0±0.6** | **81.4±1.5** | 98.6±0.2 | **90.7±1.7** | **96.7±3.2** | 72.5±5.0 |
| 0.4 | Full finetuning | 94.3 | 82.4 | 94.7 | 83.4 | 98.5 | 91.4 | 97.6 | 80.0 |
| | First layer | 88.1 | 71.2 | 91.5 | 75.5 | 98.2 | 91.0 | 93.9 | 62.4 |
| | Last layer | 86.9 | 73.4 | 92.2 | 76.4 | 98.3 | 90.6 | 96.6 | 72.1 |
| | Auto-RGN | 90.7 | 75.3 | 94.5 | 81.1 | 98.5 | 90.6 | 97.8 | 81.2 |
| | Set encoding | 94.2 | **85.3** | 93.3 | 82.0 | 98.5 | **95.1** | 97.5 | 84.2 |
| | REVO-Tune (Bin) | 93.0±0.7 | 81.1±0.5 | **96.4±0.3** | **84.2±0.5** | 98.5±0.4 | 91.4±1.2 | 97.9±0.2 | 83.0±1.2 |
| | REVO-Tune (Cont.) | **94.8±0.8** | 84.3±0.3 | 95.3±0.4 | 83.0±0.9 | **98.8±0.2** | 94.3±1.3 | **98.0±0.2** | **84.2±1.2** |
| 0.6 | Full finetuning | 93.5 | 82.7 | 95.2 | 85.8 | 98.4 | 89.4 | 99.2 | 83.1 |
| | First layer | 89.5 | 71.8 | 94.6 | 81.7 | 98.7 | 93.5 | 97.6 | 72.3 |
| | Last layer | 89.3 | 73.7 | 93.2 | 77.2 | 97.2 | 88.6 | 98.1 | 77.1 |
| | Auto-RGN | 93.1 | 80.1 | 95.4 | 82.1 | 97.5 | 89.4 | 98.1 | 83.1 |
| | Set encoding | 93.5 | 84.0 | 95.7 | 85.9 | 98.6 | 94.3 | 99.4 | 85.5 |
| | REVO-Tune (Bin) | 95.0±0.2 | 84.0±1.2 | **96.7±1.5** | 86.5±1.9 | 98.8±0.1 | 92.7±1.3 | **99.5±0.3** | 86.7±2.0 |
| | REVO-Tune (Cont.) | **95.9±2.6** | **88.5±6.5** | 96.5±1.2 | **87.3±1.1** | **99.4±0.2** | 94.3±1.4 | **99.5±0.1** | 86.7±2.0 |

freezing decisions significantly improve fine-tuning effectiveness. The advantages of REVO-Tune are particularly pronounced in low-data regimes. On the BUSI dataset with only 5% training data, continuous representation achieves 86.1% AUC and 72.3% accuracy with ResNet-18, substantially outperforming full fine-tuning (81.2% AUC, 67.0% accuracy). This trend is consistent across all datasets, highlighting the robustness of the proposed approach in data-scarce scenarios.

Our experiments reveal that training data size significantly influences the performance of fine-tuning strategy. As training data increases, the performance gap between REVO-Tune and baselines narrows, aligning with expectations that models adapt more effectively with sufficient data. Nevertheless, despite larger training sizes (e.g., 60%), REVO-Tune with continuous representation maintains its advantage. The Leaves dataset using ResNet-18 with 60% training data achieves 99.4% AUC and 94.3% accuracy versus 98.4% and 89.4% for full fine-tuning, showing that optimizing layer-wise learning rates benefits even data-rich scenarios.

Further, the proposed method demonstrates robustness across neural architectures, with continuous representation consistently outperforming baselines on both ResNet-18 and ResNet-50. Performance gains are more pronounced with the deeper ResNet-50, particularly in low-data settings. On the SRSMAS dataset with 5% training data, continuous representation achieves 87.6% AUC and 50.6% accuracy using ResNet-50, compared to 86.3% and 37.0% for full fine-tuning, indicating effective handling of deeper models with limited data.

Compared to the set encoding approach Shen et al. (2021), which applies evolutionary algorithms to optimize learning rates per layer, REVO-Tune performs comparably or slightly better across most datasets and training sizes. Our approach's advantage lies in its simplicity and computational efficiency: directly encoding learning rates as continuous or binary values reduces search space complexity compared to managing multiple learning rates per layer, enabling faster convergence and reduced computational overhead during evolutionary optimization.

Table 2: ResNet–50 results across BUSI Al-Dhabyani et al. (2020), HAM10000 Tschandl et al. (2018), Leaves Rauf et al. (2019), and SRSMAS Gómez-Ríos et al. (2019). For REVO–Tune we report mean ± std over 3 runs. Baselines include full, first, and last layer fine-tuning, Auto-RGN Lee et al. (2022), and Set encoding Shen et al. (2021). Best per column is in **bold**.

| Data size | Method | BUSI | | HAM10000 | | Leaves | | SRSMAS | |
|---|---|---|---|---|---|---|---|---|---|
| | | AUC | Acc | AUC | Acc | AUC | Acc | AUC | Acc |
| 0.05 | Full finetuning | 76.2 | 63.9 | 82.4 | 73.5 | 93.2 | 70.7 | 86.3 | 37.0 |
| | First layer | 61.5 | 56.6 | 82.3 | 69.3 | 93.0 | 71.4 | 82.2 | 28.2 |
| | Last layer | 62.7 | 56.6 | 84.1 | 70.6 | 91.3 | 63.3 | 82.2 | 26.9 |
| | Auto-RGN | 59.1 | 55.4 | 79.1 | 69.6 | 87.6 | 59.4 | 81.7 | 21.1 |
| | Set encoding | 84.2 | 68.7 | 85.4 | 75.8 | 95.7 | 81.7 | 82.0 | 38.0 |
| | **REVO-Tune (Bin)** | 71.1±4.2 | 58.3±4.9 | **88.3±0.6** | **76.5±0.7** | 92.8±3.1 | 66.4±1.9 | 83.5±2.9 | 30.8±2.2 |
| | **REVO-Tune (Cont.)** | **86.3±4.3** | **71.3±0.3** | 87.9±1.7 | 76.6±1.7 | **95.8±0.9** | 76.6±0.5 | **87.6±0.8** | **50.6±0.7** |
| 0.2 | Full finetuning | 89.1 | 74.6 | 91.7 | 77.8 | 96.2 | 81.5 | 95.9 | 66.4 |
| | First layer | 73.6 | 60.9 | 92.2 | 76.9 | 97.5 | 83.9 | 93.9 | 59.1 |
| | Last layer | 80.3 | 65.4 | 90.4 | 75.1 | 96.5 | 81.7 | 95.0 | 62.3 |
| | Auto-RGN | 69.0 | 55.1 | 90.7 | 75.1 | 95.9 | 68.7 | 93.9 | 53.8 |
| | Set encoding | 90.4 | 76.7 | 93.9 | 80.6 | 98.0 | 88.8 | 97.0 | 76.1 |
| | REVO-Tune (Bin) | 88.8±0.6 | 73.7±1.4 | 93.3±0.8 | 81.7±0.3 | 97.9±0.4 | 82.8±2.0 | 96.3±0.6 | 73.3±3.3 |
| | REVO-Tune (Cont.) | **92.6±0.5** | **81.0±1.6** | **94.5±0.2** | **81.5±0.5** | **98.8±0.3** | **89.9±0.8** | **97.4±0.3** | **76.9±0.8** |
| 0.4 | Full finetuning | 91.5 | 76.9 | 94.0 | 83.3 | 98.1 | 90.6 | 98.0 | 86.1 |
| | First layer | 83.8 | 65.7 | 94.6 | 80.4 | 98.3 | 89.0 | 96.9 | 77.6 |
| | Last layer | 86.6 | 71.5 | 92.4 | 77.3 | 97.4 | 82.4 | 97.5 | 75.8 |
| | Auto-RGN | 78.9 | 55.1 | 92.8 | 77.9 | 97.6 | 86.1 | 96.9 | 77.0 |
| | Set encoding | 92.6 | 80.8 | 96.6 | 86.1 | 97.6 | 88.6 | 97.0 | 82.4 |
| | REVO-Tune (Bin) | 93.6±0.3 | 79.8±0.4 | **95.5±0.4** | **85.4±0.6** | 99.0±0.2 | 92.7±1.3 | 98.0±0.4 | 84.8±1.7 |
| | REVO-Tune (Cont.) | **94.4±0.2** | **84.9±0.8** | 95.4±0.2 | 83.8±0.3 | 98.9±0.4 | 93.5±1.1 | 98.1±0.7 | 87.9±2.9 |
| 0.6 | Full finetuning | 92.6 | 83.3 | 95.8 | 84.2 | 97.2 | 92.7 | 99.5 | 89.2 |
| | First layer | 87.3 | 75.6 | 95.0 | 82.0 | 97.9 | 87.8 | 97.5 | 79.5 |
| | Last layer | 87.4 | 71.8 | 93.1 | 79.4 | 98.2 | 86.2 | 97.9 | 79.5 |
| | Auto-RGN | 85.4 | 69.2 | 93.9 | 80.9 | 96.5 | 77.2 | 97.9 | 80.7 |
| | Set encoding | 92.7 | 82.1 | 96.5 | 87.1 | 98.9 | 91.9 | 96.2 | 77.1 |
| | REVO-Tune (Bin) | 93.7±0.5 | 84.6±2.0 | **96.4±0.2** | **86.6±0.3** | 98.9±0.3 | 94.3±1.3 | 98.7±0.2 | 86.7±1.8 |
| | REVO-Tune (Cont.) | **95.3±0.5** | **86.5±1.6** | 95.9±0.1 | 84.9±0.3 | **99.0±0.7** | **95.1±3.3** | **99.4±0.3** | **91.6±0.6** |

## 5 CONCLUSIONS AND FUTURE WORKS

We present REVO-Tune, which applies CMA-ES evolutionary optimization to discover layer-specific learning rates for neural network fine-tuning automatically. Our approach employs binary and continuous encoding strategies to optimize layer-wise configurations without gradient-based hyperparameter search. Experiments across four datasets demonstrate consistent improvements over traditional fine-tuning, achieving 2-4% accuracy and 1-3% AUC gains. The method proves particularly effective in data-scarce scenarios where gradient-based optimization becomes unreliable. Our analysis reveals that evolutionary optimization can automatically discover dataset-specific learning rate patterns, with continuous encoding excelling in performance-critical scenarios and binary encoding providing computational efficiency.

Future work will explore comparisons with modern hyperparameter optimization methods and evaluation on larger-scale datasets to validate the approach's effectiveness further.

REPRODUCIBILITY STATEMENT

We have made extensive efforts to ensure the reproducibility of our work. We provide full experimental details in Section 4.1, including datasets, architectures (ResNet-18/50), evaluation metrics, and CMA-MV parameters (population 10, 10 iterations, global LR 1e-3 for binary, [0,1] range for continuous). Baselines were tuned with identical budgets using HyperOpt and successive halving. Results are reported as mean ± std over three runs, with weight update rules in Eqs. 1–2. Each run required 6 hours on an NVIDIA RTX A5000. Large language models (LLMs) were used only for language polishing; all ideas, methods, and experiments are by the authors. Source code will be released upon acceptance.

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

# Supplementary Material

TABLE OF CONTENTS

# A ABLATION STUDIES

## A.1 LAYER-WISE LEARNING RATE PATTERNS

Figures 3 and 4 visualize the magnitudes of the optimized learning rates across datasets and training data ratios to gain insight into the layerwise learning rate patterns discovered by the proposed evolutionary algorithm. Figure 3 reveals distinct patterns in the learning rate distributions across datasets. For instance, in the HAM10000 dataset, the algorithm assigns lower learning rates to earlier layers, suggesting the importance of adapting low-level features for this task. In contrast, the SRSMAS dataset has a more uniform distribution of learning rates, indicating the need for balanced adaptation across all layers.

Figure 4 illustrates the influence of the training data size on the learning rate patterns. As the proportion of training data increases, the algorithm tends to assign higher learning rates to larger layers. This observation aligns with the expectation that more training data allow the model to adapt a more significant portion of the parameters to the target domain. These visualizations highlight the ability of the evolutionary algorithm to discover dataset-specific and data-size-dependent learning rate patterns, enabling effective fine-tuning adaptation.

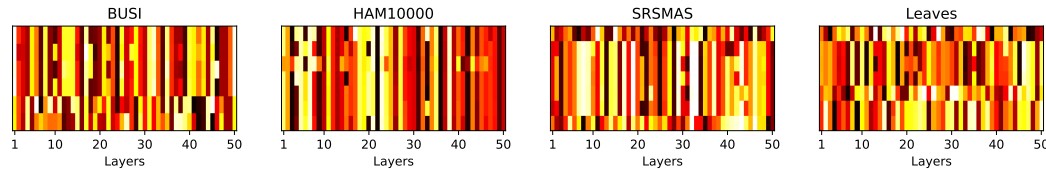

Figure 3: Distribution of the learning rate magnitudes for each dataset

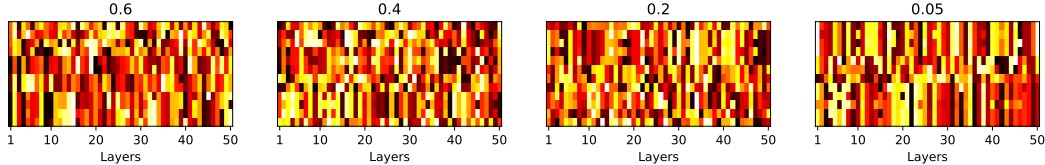

Figure 4: Distribution of the learning rate magnitudes for different training data ratios

## A.2 PERFORMANCE COMPARISON ON DIFFERENT MODELS

This study presents an ablation study conducted using various well-known DNN architectures beyond the ResNet models evaluated in the principal experiments to validate the effectiveness of the proposed layerwise learning rate optimization approach. Specifically, ConvNext Liu et al. (2022a), EfficientNet Tan & Le (2019), MobileNet Howard et al. (2017), and ShuffleNet Zhang et al. (2018) were evaluated on the target dataset. Table 3 compares the proposed method and the full fine-tuning baseline across these diverse model architectures. Classification accuracy and AUC metrics provide a comprehensive evaluation.

Table 3: We compare the Accuracy and AUC of different models using Full finetuning fine-tuning and the proposed method. We search for the best learning rate for the Full finetuning method using Ray tuner Liaw et al. (2018)

|  | Full finetuning fine-tuning | | REVO-Tune (Binary Rep.) | | REVO-Tune (Cont. Rep.) | |
| --- | --- | --- | --- | --- | --- | --- |
|  | Accuracy | AUC | Accuracy | AUC | Accuracy | AUC |
| ConvNext | 0.777 | 0.9846 | 0.8667 | 0.9865 | 0.8424 | 0.9849 |
| EfficientNet | 0.753 | 0.9705 | 0.7697 | 0.9765 | 0.8 | 0.9773 |
| MobileNet | 0.7407 | 0.9611 | 0.8182 | 0.9778 | 0.8424 | 0.9835 |
| ShuffleNet | 0.666 | 0.9327 | 0.7576 | 0.9746 | 0.7394 | 0.9636 |

The proposed layerwise learning rate optimization method consistently outperformed the full fine-tuning approach across all considered models. For instance, with the ConvNext architecture, the proposed method achieved an accuracy of 84.24%, surpassing the full fine-tuning baseline by a substantial 6.64 percentage points. Similar gains were observed for EfficientNet and MobileNet, with improved accuracy of 4.7% and 10.17%, respectively. Even the ShuffleNet architecture, which performed worse overall compared to the other models, saw a notable boost of 7.34 percentage points in accuracy using the proposed method over the full fine-tuning approach. These consistent improvements across diverse model families substantiate the general applicability and effectiveness of the layerwise optimization strategy.

### A.3 COMPUTATIONAL EFFICIENCY ANALYSIS

While more computationally intensive than standard fine-tuning, the proposed evolutionary fine-tuning approach offers a favorable trade-off between computational cost and optimization efficacy. The proposed method evaluates 100 distinct configurations, comparable to a grid search over five learning rates, five weight decay values, and four epoch numbers. However, the proposed approach adaptively refines its search space, concentrating the computational resources on promising regions of the hyperparameter space and enabling layer-specific learning rate optimization, which is infeasible with a traditional grid search.

The computational complexity of the algorithm scales linearly with the number of network layers, making it suitable for modern deep architectures. In contrast, exhaustive grid search methods exponentially grow in complexity as the number of hyperparameters increases, becoming prohibitively expensive for fine-grained layerwise optimization. The population-based nature of the proposed algorithm facilitates efficient parallelization in multi–graphics processing units or distributed computing environments, potentially reducing wall-clock time and offsetting increased computational demands. Despite requiring more computation than a single fine-tuning run, the performance gains observed across datasets and architectures suggest that this additional computational investment yields substantial improvements in model adaptation, particularly in scenarios with limited training data.

### B COMPUTATIONAL RESOURCES

Each experimental run could be completed in approximately 6 hours using an NVIDIA RTX A5000 with 24GB GPU memory.

