# OpenReview forum: "Evolve to Adapt, Not Guess: A Gradient-Free and Robust Framework for Layer-Wise Fine-Tuning via Evolutionary Learning Rate Optimization"
_ICLR.cc/2026/Conference — ICLR 2026 Conference Withdrawn Submission_

### Official Review · Reviewer_pXKd · 2025-10-31

**Soundness:** 3
**Presentation:** 3
**Contribution:** 2
**Rating:** 4
**Confidence:** 2

**Summary:**

This paper presents REVO-Tune, a gradient-free fine-tuning framework that employs evolutionary optimization (CMA-ES) to automatically determine layer-wise learning rates for pretrained models. The approach introduces two encoding strategies (i.e., binary and continuous) and achieves consistent performance gains across multiple datasets, especially in low-resource settings.

**Strengths:**

1) By leveraging CMA-ES, the proposed framework eliminates dependence on noisy gradients, making it particularly effective in low-data or noise-prone domains.

2) The method tackles the practical challenge of determining layer-wise learning rates for fine-tuning through a novel evolutionary optimization approach.

3) The framework consistently enhances performance across diverse task scenarios.

**Weaknesses:**

1) All experiments are conducted on image classification tasks. The paper lacks evaluations on modern architectures (e.g., ViT) or other modalities (e.g., NLP, multimodal tasks), which limits the generality of its claims.

2) While CMA-ES is known to be computationally expensive, the paper only provides a brief runtime report without including convergence curves or detailed comparisons of computational efficiency with grid or random search. Adding an analysis of the algorithm’s convergence behavior—such as performance improvement per iteration or generation—would greatly strengthen the paper and clarify its practical feasibility.

3) In line 774, the authors state that “the computational complexity of the algorithm scales linearly with the number of network layers.” However, from the perspective of evolutionary search, the search space typically expands exponentially as the number of layers increases. Could the authors clarify how the proposed method maintains linear computational complexity under such conditions?

**Questions:**

1) The experiments in the paper are limited to image classification tasks. Could the authors comment on the generality of their approach and whether it has been evaluated or could be applied to modern architectures, such as Vision Transformers (ViTs), or to other modalities, such as NLP or multimodal tasks?

2) Could the authors provide more detailed analyses, such as convergence curves or comparisons of computational efficiency with grid or random search, to clarify the practical feasibility of the algorithm?

3) Could the authors clarify how the proposed method maintains linear computational complexity under the conditions described in line 774?

---

### Official Review · Reviewer_XTen · 2025-10-31

**Soundness:** 3
**Presentation:** 3
**Contribution:** 2
**Rating:** 4
**Confidence:** 2

**Summary:**

This paper proposed a new method for determining optimal layer-specific learning rates for fine-tuning. Their method  (REVO-Tune) uses evolutionary optimization, gradient-free search, and is able to boost accuracy by 2-4% and AUC by 1-3% compared to other well-used methods. The approach is particularly well-suited for data-scarce environments where traditional gradient-based hyperparameter tuning isnt as reliable.

**Strengths:**

I'm not as familiar with methods in evolutionary optimization but this is an interesting paper and is well-written. Some strengths:
 - one of the primarily strengths of REVO-Tune is its efficacy in data-scarce fine-tuning situations. This is because the evolutionary algorithms do not rely on gradients which can have high variance and are unreliable with few examples. REVO-Tune uses a black box evolutionary search that relies on final validation performance and is much more robust to noise.
 - Another interesting feature of the REVO-Tune algo is the capability for granular layer-specific tuning. REVO-Tune does this by determining layer-wise configurations and assigns a learning rate ot each layer. They show their method is competitive with other baselines across multiple datasets.

**Weaknesses:**

I"m not as familiar with evolutionary optimization. My proposed weaknesses are contained in the questions outlined below, mostly around the positioning of REVO-Tune as "gradient free". See below.

Also, the abstract states that the "Code is provided in the supplementary material" but here is no link to any code.

**Questions:**

I'm not an expert on evolutionary optimization so most of my questions are concerning these techniques and methods.

It is stated that one of the main advantages of REVO-Tune is that it's gradient free. Is this fully accurate? The evolutionary optimization via CMA-ES does involve gradients, correct? Also, doesn't the actual model fine-tuning performed at every eval step involve gradients? Can you clarify this in the paper. This seems more like a hybrid approach; ie a gradient free hyperparameter search wrapping a gradient based finetuning trainer. Is that right?

One of the stated advantages of this method is that it performs well when data is scarce and its usefulness in practical applications. However, the evolutionary search itself can be very expensive and requires 100 full finetuning evaluations to find the best learning rate with a population of 10. This is an important trade-off that needs to be highlighted. Sure this is better than grid-search but how does this compute compare to standard fine-tuning methods? My guess is this is much more computationally expensive. Is that correct?

---

### Official Review · Reviewer_M8TA · 2025-11-01

**Soundness:** 2
**Presentation:** 3
**Contribution:** 2
**Rating:** 2
**Confidence:** 4

**Summary:**

The paper proposes REVO-Tune, a framework using evolutionary optimization (CMA-ES) to automatically find layer-specific learning rates during fine-tuning. It aims to replace manual heuristics, particularly in data-scarce situations where gradient-based hyperparameter tuning is unstable. The method employs binary (freeze/train) and continuous (per-layer rate) encoding strategies. The primary evaluation is conducted on ResNet architectures across four datasets.

**Strengths:**

- Problem Motivation: The paper addresses a valid real-world issue: the inconsistency of standard fine-tuning strategies (like first-layer vs last-layer) across different datasets, as effectively illustrated in Figure 1.

- Exploration of Encodings: The comparison between binary and continuous evolutionary encodings provides some insight into the trade-offs between search space complexity and fine-grained control.

**Weaknesses:**

- Insufficient Architectural Evaluation: The main body of the paper relies entirely on outdated ResNet-18 and ResNet-50 architectures. While an ablation study in Appendix A.2 includes other models (ConvNext, EfficientNet, etc.), it is methodologically flawed as it only compares REVO-Tune against basic "Full fine-tuning" . Stronger baselines used in the main text, such as Auto-RGN, are completely omitted from this appendix, failing to establish if the method is competitive on modern architectures.

- Marginal Improvements in Target Scenarios: The paper claims robustness in data-scarce scenarios, but the results are often marginal and inconsistent. Only the std of the proposed method is reported. Given the marginal improvement, statistical tests are necessary.

- High Computational Cost for Low Gain: The proposed method is computationally expensive, requiring approximately 6 hours per run on an RTX A5000 due to the population-based evolutionary search (100 total evaluations). Given the often marginal performance gains noted above, the cost-benefit ratio of this framework is poor compared to standard methods. Besides, to ensure fair comparison, other method should be allowed for the same amount of budget for hyperparameter searching.

**Questions:**

1. Why were key baselines like Auto-RGN omitted from the architectural ablation study in Appendix A.2? Without them, it is difficult to assess the true competitiveness of REVO-Tune on these models.

2. Can you address the inconsistency in data-scarce performance? For instance, why does the method fail to provide meaningful AUC gains (and lose accuracy) on SRSMAS at 5% data compared to simple full fine-tuning, despite this being the exact scenario deemed suitable for the framework?

3. Given the 6-hour runtime, did you compare this against running a standard random search or Bayesian optimization for simple hyperparameters (e.g., one global learning rate and one weight decay) for the same 6-hour duration?

---

### Official Review · Reviewer_DWyU · 2025-11-01

**Soundness:** 1
**Presentation:** 3
**Contribution:** 1
**Rating:** 2
**Confidence:** 4

**Summary:**

The paper proposes REVO-Tune, an evolutionary optimization approach for automatically discovering optimal layer-specific learning rates during neural network fine-tuning. The approach uses two types of encodings (binary, for determining which layers to freeze, and continuous, to set the learning rate for the active layers), and combines them with a CMA-ES based search. The authors claim that this approach is more efficient and thus able to achieve higher performance on data-scarce settings.

**Strengths:**

1) The proposed approach is well motivated and presented.
2) The evaluation consists of multiple datasets and baselines.
3) The paper is well organized.

**Weaknesses:**

1) Limited evaluation - the evaluation is conducted on four relatively small datasets, all from the image domain. It would have been helpful to evaluate the approach on datasets from several domains.

2) Lack of relevant baselines - the authors compare their approach to simple heuristics and two methods from 2021 and 2022. The authors do not compare their approach to more recent work, and to several types of highly effective and popular approaches, including:
a) Bayesian optimization
b) AutoML-based frameworks (e.g., Optuna, Auto-SKLearn)
c) Newer variants of Auto-RGN

3) The chosen training budget seems a little arbitrary. Why use 100? At the minimum, one would be expected to evaluate a range of budgets.

4) There is no analysis of the proposed methods - in which circumstances does it do better? at which worse? are there any insights regarding dataset or network characteristics that affect its performance?

**Questions:**

I would like the authors to respond to the weaknesses listed above.

---

### Note · Authors · 2025-11-14

I have read and agree with the venue's withdrawal policy on behalf of myself and my co-authors.